# Rapid Green Extractions of C-Phycocyanin from *Arthrospira maxima* for Functional Applications

**Mari Carmen Ruiz-Domínguez [1,\*]**, **Marjorie Jáuregui [1]**, **Elena Medina [1]**, **Carolina Jaime [2]** and **Pedro Cerezal [1]**

[1]  Laboratorio de Compuestos Bioactivos (LAMICBA), Departamento de Ciencias de los Alimentos y Nutrición, Facultad de Ciencias de la Salud, Universidad de Antofagasta, Avda. Universidad de Antofagasta 02800, Antofagasta 1240000, Chile; marjorie.jauregui@uantof.cl (M.J.); e-b-medina@hotmail.com (E.M.); pedro.cerezal@uantof.cl (P.C.)

[2]  Atacama Bio Natural Products S.A., Vía 5 Esq. Vía 9, Bajo Molle, Iquique 1100000, Chile; cjaime@atacamabionatural.com

\*  Correspondence: maria.ruiz@uantof.cl; Tel.: +56-552-633-660

**Abstract:** Cyanobacteria are a rich source of bioactive compounds, mainly in the *Arthospira* sp., and one of the most interesting components in recent years has been C-phycocyanin (C-PC). There have been several conventional methods for their extraction, among which stand out: chemical products, freezing-thawing (FT); enzymatic, and maceration (M); which have come to be replaced by more environmentally friendly methods, such as those assisted by microwaves (MW) and high-pressure homogenization (HPH). The aim of the research was to use these two "green extraction processes" to obtain C-PC from cyanobacteria *Arthrospira maxima* because they improve functionality and are fast. Extractions of C-PC were studied by means of two experimental designs for MW and HPH, based on a response surface methodology (RSM) employing, firstly, a factorial design $3^3$: power (100, 200, and 300 W), time (15, 30, and 60 s), and types of solvents (distiller water, Na-phosphate buffer and, distiller water: Na-phosphate buffer (Ph 7.0; 1:1, *v/v*); and secondly, two factors with different levels: Pressure (800, 1000, 1200, 1400, and 1600 bar) and, types of solvents (distilled water, Na-phosphate buffer (pH 7.0) 100 mM and, Na-phosphate buffer:water 1:1, (*v/v*)). Optimum C-PC content was achieved with the HPH process under Na-phosphate solvent at 1400 bar (291.9 ± 6.7 mg/g) and the MW method showed improved results using distilled water as a solvent at 100 W for 30 s (215.0 ± 5.5 mg/g). In the case of conventional methods, the freeze–thawing procedure reached better results than maceration using the buffer (225.6 ± 2.6 mg/g). This last one also did not show a significant difference between solvents (a range of 147.7–162.0 mg/g). Finally, the main advantage of using green extractions are the high C-PC yield achieved, effectively reducing both processing times, costs, and increasing the economic and functional applications of the bioactive compound.

**Keywords:** cyanobacteria; phycobiliproteins; microwave; high-pressure homogenization; functional ingredient

## 1. Introduction

In the present, many reports have focused on the prospect of potential functional ingredients due to an increase of interest in a healthy lifestyle [1–3]. The functional ingredients are bioactive compound(s) isolated or purified from natural sources with health-promoting properties demonstrating their relevant role in disease risk reduction [3,4]. Phycobiliproteins (PBP) are a group of coloured proteins mainly present in cyanobacteria (blue-green algae) and red algae. They are components of photosynthetic light-harvesting antenna complexes and they have a broad spectrum of functions [5–7].

In general, these proteins are classified into two large groups based on their colours and absorption spectra such as phycoerythrin (PE, red − λmax = 490–570 nm) and phycocyanin (PC, blue − λmax = 610–625 nm). In particular, the phycocyanins are included: C-phycocyanin (C-PC), R-phycocyanin (R-PC) and allophycocyanin (APC) which differ in their spectral properties, structural composition, and colour [8–10]. Nowadays, PBP has a broad market in pharmaceutical, biomedical, food and cosmetic industries due to their varied beneficial properties [5,11]. The main use of PBP described is as non-toxic and non-carcinogenic natural food colorants [11,12] and they exhibit a consolidated market in the clinical and immunological area due to their fluorescent capacity [13]. PC has also shown anti-inflammatory, antiplatelet, anti-cancer, nephroprotective, and hepatoprotective properties justified by its antioxidant activity [14,15]. Moreover, the current market of PBP is assessed to be more than 60 million USD [16] highlighting the role of C-PC as a fluorescent agent where its commercial value depends on its degree of purity (food-grade or analytical grade) [17].

The most named cyanobacteria for producing C-PC as natural blue pigment is *Arthrospira*, also known as Spirulina, a blue-green alga [13]. Specifically, *Arthrospira plantesis* and *Arthrospira maxima* are the species mostly used with a wide market such as food additives, health food, cosmetics, pharmaceuticals and medicine [18–21]. For those reasons, *Arthrospira* sp. is a promising source of C-PC that represents an opportunity for economic profit.

Many reports have focused on improving the culture conditions from physical and chemical to engineer's parameters of the process, e.g., optimization of culture media, selection of stress conditions, bioreactor design etc. for increasing the bioactive compound synthesis in microalgae and cyanobacteria [22,23]. However, the extraction methods are also a very important factor for the maximum recovery of PBP (functional ingredients). In the case of conventional methods such as maceration extraction (M, soaking in solvent), freezing and thawing are the most used for PBP extraction; however, the process can be quite lengthy [12,24]. Normally, these types of extraction methods start from dry biomass and then a thermal extraction with organic or aqueous solvents is used, depending on the polarity of the target compounds to be extracted [25]. The main problems presented in the conventional extraction process are moderate extraction efficiency and selectivity and high solvent consumption [25]. On the other hand, there are green extraction techniques that are defined because they do not use solvents or chemicals harmful to the environment and it is also reported that they show better results for the extraction process. They could be classified in general as: pressurised liquid extraction (PLE), supercritical fluid extraction (SFE), microwave extraction (MW), ultrasound (US) and high-pressure homogenization (HPH) between other technologies [25]. Additionally, the PBP are also soluble-water compounds, and for this reason the selection of suitable solvent as buffer is also a very crucial factor for obtaining high-yield and -quality PBP [26]. Thereby, we evaluate in this work the effect of two green extraction techniques as MW and HPH and three solvents (distilled water, phosphate buffer, and phosphate buffer:water (1:1 *v/v*)) versus conventional extraction in C-PC content from the cyanobacteria *Arthrospira maxima*. Finally, this study could aim to strengthen the rapid extraction process of C-PC by easing the multiple applications due to the need for bioactive compounds and even improve the times and costs of processing.

## 2. Materials and Methods

### 2.1. Material

*Arthrospira maxima* was the cyanobacteria selected in this research. The Chilean company Atacama Bio Natural Products Inc. (Iquique, Chile) kindly donated the samples to us. The cyanobacteria was cultured under control condition and was harvested in its exponential growth phase. Then, the biomass was dried under an industrial spray-drying system (Galaxy Spray drying model 3530, Buenos Aires, Argentina) and was worked as a spray-dried powder in this study. The biomass was stored in vacuum until use. Three independent solvents were used to extract C-PC from *A. maxima*

being: (i) distilled water; (ii) Na-Phosphate buffer pH 7.0, 100 mM, and (iii) Na-Phosphate buffer pH 7.0, 100 mM, water 1:1 (*v/v*) and all were analytical grades.

## 2.2. Conventional Extraction for Phycobiliproteins Extraction

### 2.2.1. Maceration Extraction

For extraction, 20 mg of cyanobacterial dry biomass with 5 mL of the independent solvents were used and subjected to repeated M extraction and vortex (Velp Scientifica Mixer-Wizard, Usmate, Italy) to mix it. The cell debris was removed by centrifugation (Champion S-50D, Vernon Hills, IL, USA) at $3140\times g$ for 5 min. The supernatant was pooled and labeled as crude extract and the extraction solvent was colourless. This was performed in triplicate.

### 2.2.2. Freeze–Thawing Extraction

The cyanobacterial dry biomass (20 mg per sample) was resuspended in 5 mL of the independent solvents and subjected to repeated freeze–thawing (Freezer DAEWOO FR385S, Gangnam-gu, Seoul, Korea) cycles of −22 °C and 20 °C temperature shocks for the release of PBP. The cell debris was removed by centrifugation at $3140\times g$ for 5 min. The supernatant was pooled and labeled as crude extract and extraction solvent was colourless. This was performed in triplicate.

## 2.3. Green Extraction Design for Phycobiliproteins Extraction

### 2.3.1. Microwave (MW) Design

Microwave process was performed using a domestic microwave oven (Fensa MF-28G model, Valparaíso, Chile) which is capable of operating at a maximum input of 1000 W at a frequency of 2450 MHz; 20 mg of *A. maxima* were used with 5 mL of independent solvents performing several extractions and the supernatant was colorless. Then the aliquots were centrifuged at $3140\times g$ for 5 min. To optimize MW extraction conditions, a factorial experimental design $3^3$ was used based on three factors with 3 levels, these being: power (100, 200 and 300 W), time (15, 30 and 60 s) and solvent (distilled water, Na-phosphate buffer (pH 7.0) 100 mM and Na-phosphate buffer:water 1:1, (*v/v*)). The effect of the factors and its interactions on specific response as C-PC content was studied. A total of 81 experiments (n = 3) as shown in Table 1 for *A. maxima*.

**Table 1.** Experimental design matrix including extraction conditions and results for C-phycocyanin (C-PC) content as response variable studied for the optimization of the microwaving (MW) of the cyanobacteria *Arthrospira maxima*. Values presented are mean ± standard deviation (SD), n = 3. Power (100 (−1), 200 (0) and 300 (1), W), Time (15 (−1), 30 (0) and 60 (1), s) and Solvent Type (water (−1), buffer (0) and buffer:water (1)).

| | MW Extraction Conditions | | | | | C-PC Content (mg/g) |
|---|---|---|---|---|---|---|
| Exp. | Power (W) | | Time (s) | | Solvent Type * | |
| 1 | −1 | (100) | −1 | (15) | −1 | (water) | 201.9 ± 5.1 |
| 2 | −1 | | −1 | | 0 | (Buffer) | 148.9 ± 2.0 |
| 3 | −1 | | −1 | | +1 | (Buffer:water) | 149.1 ± 5.6 |
| 4 | −1 | | 0 | (30) | −1 | | 215.0 ± 5.5 |
| 5 | −1 | | 0 | | 0 | | 161.5 ± 6.9 |
| 6 | −1 | | 0 | | +1 | | 165.9 ± 9.3 |
| 7 | −1 | | +1 | (60) | −1 | | 169.5 ± 3.8 |
| 8 | −1 | | +1 | | 0 | | 160.9 ± 8.4 |
| 9 | −1 | | +1 | | +1 | | 168.6 ± 12.2 |
| 10 | 0 | (200) | −1 | | −1 | | 192.5 ± 1.3 |
| 11 | 0 | | −1 | | 0 | | 170.7 ± 7.5 |
| 12 | 0 | | −1 | | +1 | | 143.8 ± 16.9 |

**Table 1.** *Cont.*

| Exp. | MW Extraction Conditions | | | C-PC Content (mg/g) |
| | Power (W) | Time (s) | Solvent Type * | |
|---|---|---|---|---|
| 13 | 0 | 0 | −1 | 134.7 ± 2.6 |
| 14 | 0 | 0 | 0 | 121.2 ± 24.9 |
| 15 | 0 | 0 | +1 | 136.9 ± 1.8 |
| 16 | 0 | +1 | −1 | 153.0 ± 5.2 |
| 17 | 0 | +1 | 0 | 87.5 ± 17.4 |
| 18 | 0 | +1 | +1 | 172.9 ± 16.1 |
| 19 | +1 (300) | −1 | −1 | 109.1 ± 12.7 |
| 20 | +1 | −1 | 0 | 61.8 ± 25.9 |
| 21 | +1 | −1 | +1 | 92.0 ± 8.1 |
| 22 | +1 | 0 | −1 | 85.7 ± 30.0 |
| 23 | +1 | 0 | 0 | 46.8 ± 14.2 |
| 24 | +1 | 0 | +1 | 86.9 ± 10.7 |
| 25 | +1 | +1 | −1 | 37.8 ± 6.6 |
| 26 | +1 | +1 | 0 | 35.1 ± 5.7 |
| 27 | +1 | +1 | +1 | 44.2 ± 2.2 |

* Three independent extraction solvents were used: (−1) water: distilled water; (0) Buffer: Na-Phosphate buffer pH 7.0, 100 mM and (+1) Buffer:water: Na-Phosphate buffer pH 7.0, 100 mM:water 1:1 (*v/v*).

### 2.3.2. High-Pressure Homogenization (HPH) Design

A GEA Niro Soavi Homogenizer (Panda PLUS model, Parma, Italy) was used as the HPH system. The ratio of *A. maxima* biomass and extraction solvent used was 2:100 (*w/v*) with two passages. After the HPH process, the aliquots were centrifuged at 3140× *g* for 5 min to quantify the C-PC. The factors evaluated were the pressure applied in the homogenization and solvents. For optimizing HPH extraction conditions, a factorial experimental design was used based on two factors, considering five levels of pressure: 800, 1000, 1200, 1400 and, 1600 bar. However, in the factor solvent there were three levels: distilled water, Na-phosphate buffer (pH 7.0) 100 mM and, Na-phosphate buffer:water 1:1, (*v/v*). The effect of the factors and its interactions on specific response as C-PC content was studied. A total of 15 experiments (n = 3) as shown in Table 2 for *A. maxima*.

**Table 2.** Experimental design matrix including extraction conditions and results for C-PC content as response variable studied for the optimization of the high-pressure homogenization (HPH) of the cyanobacteria *Arthrospira maxima*. Values presented are mean ± SD, n =3.

| Exp. | HPH Extraction Conditions | | | C-PC Content (mg/g) |
| | Pressure (Bar) | | Solvent * | |
|---|---|---|---|---|
| 1 | −1 (800) | −1 | (Water) | 218.1 ± 10.1 |
| 2 | −1 | 0 | (Buffer) | 225.9 ± 0.4 |
| 3 | −1 | +1 | (Buffer:water) | 179.3 ± 1.1 |
| 4 | −0.5 (1000) | −1 | | 224.0 ± 3.3 |
| 5 | −0.5 | 0 | | 233.7 ± 5.6 |
| 6 | −0.5 | +1 | | 203.4 ± 6.9 |
| 7 | 0 (1200) | −1 | | 252.5 ± 4.1 |
| 8 | 0 | 0 | | 268.8 ± 6.4 |
| 9 | 0 | +1 | | 212.3 ± 9.3 |
| 10 | +0.5 (1400) | −1 | | 249.1 ± 6.1 |
| 11 | +0.5 | 0 | | 291.9 ± 6.7 |
| 12 | +0.5 | +1 | | 210.3 ± 1.2 |
| 13 | +1 (1600) | −1 | | 224.3 ± 6.4 |
| 14 | +1 | 0 | | 257.4 ± 7.4 |
| 15 | +1 | +1 | | 177.9 ± 5.9 |

* Three independent extraction solvents were used: (−1) water: distilled water; (0) Buffer: Na-Phosphate buffer pH 7.0, 100 mM and (+1) Buffer:water: Na-Phosphate buffer pH 7.0, 100 mM:water 1:1 (*v/v*).

### 2.4. C-Phycocyanin (C-PC) Quantification

Each of the crude PBP supernatants was measured by spectrophotometer Shimadzu Ultraviolet–Visible (UV–vis) 1280 (Japan) at 620 and 652 nm to determine C-phycocyanin concentration (*w/v*) as described by Bennett and Bogorad [27]:

$$\text{C} - \text{PC concentration (mg/mL)} = \frac{[(A620 - 0.474 \cdot A652)]}{5.34}$$

Then, C-phycocyanin content (w/w) was calculated using this equation:

$$\text{C} - \text{PC (mg/g)} = \frac{(\text{C} - \text{PC concentration}) \cdot \text{V}}{\text{dW}}$$

where C-PC is the crude C-phycocyanin in mg/g, $A_{620}$ and $A_{652}$ represent the absorbance of C-phycocyanin at 620 and 652 nm, V is the volume of solvent used in mL and dW represents the dry biomass in grams.

### 2.5. Statistical Analysis

The experimental design and data analysis were carried out using response surface methodology (RSM) with Statgraphics Centurion XVI® (StatPoint Technologies, Inc., Warrenton, VA, USA) software. The effects of the independent factors on the response variables in the separation process were assessed using the pure error, considering a level of confidence of 95% for all the variables. The effect of each factor and its statistical significance, for each of the response variables, was analyzed from the standardized Pareto chart. The response surfaces of the respective mathematical models were also obtained, and the significances were accepted at $p \leq 0.05$. A multiple response optimization was carried out by the combination of experimental factors, looking for maximizing the desirability function for the responses in the extracts. The optimal condition to obtain the highest extraction yield was introduced by the software. The regression equation was also performed by MW and HPH techniques. All experiments were carried out in triplicate (n = 3) and the data was shown as mean ± SD (standard deviation).

## 3. Results and Discussion

### 3.1. Optimization of MW Extraction Conditions for C-PC from Arthrospira Maxima

The obtained results are summarized in Table 1. As can be seen, higher C-PC content was obtained using distilled water as solvent, a power of 100 W for 30 s. Figure 1 represents standardized Pareto charts (Figure 1A) where non-significant variables of the model were eliminated and the response surfaces on the basis of the factors selection (Figure 1B solvent was distiller water; Figure 1C power at 100 W; Figure 1D time at 30 s). The regression equation was also performed in MW technique. It was adjusted to the C-PC content as the following model (1):

$$\text{C-PC (mg/g)} = 120.02 - 51.81 \cdot \text{P} - 18.48 \cdot \text{t} - 18.51 \cdot \text{P}^2 - 14.55 \cdot \text{P} \cdot \text{t} + 7.76 \cdot \text{P} \cdot \text{S} + 11.31 \cdot \text{t} \cdot \text{S} + 27.00 \cdot \text{S}^2 \quad (1)$$

where C-CP is C-phycocyanin content in mg/g, P: is power in W, t: is the extraction time in seconds, and S: is solvent type (Table 3). This confirms what is described by Juin, et al. [28] being that the overexposure to high heat (intense power) could damage the C-PC content in the extraction. In this way, our results showed that the C-PC content reached from *A. maxima* was higher than other results described in microalgal phycobiliproteins having ~198.3 mg/g as the predicted optimal value by statistic software and ~215.0 ± 5.5 mg/g as the real extracted content. For example, İlter, et al. [29] reported several solvents for C-PC extracting from *Arthrospira platensis* where the range was from 56.2 ± 0.14 to 96.65 ± 0.15 mg/g. This study was also performed with an experimental design where

the best condition using MW extraction was at 150 W and 120 s with a value of 7.45 fold lower than our maximum content.

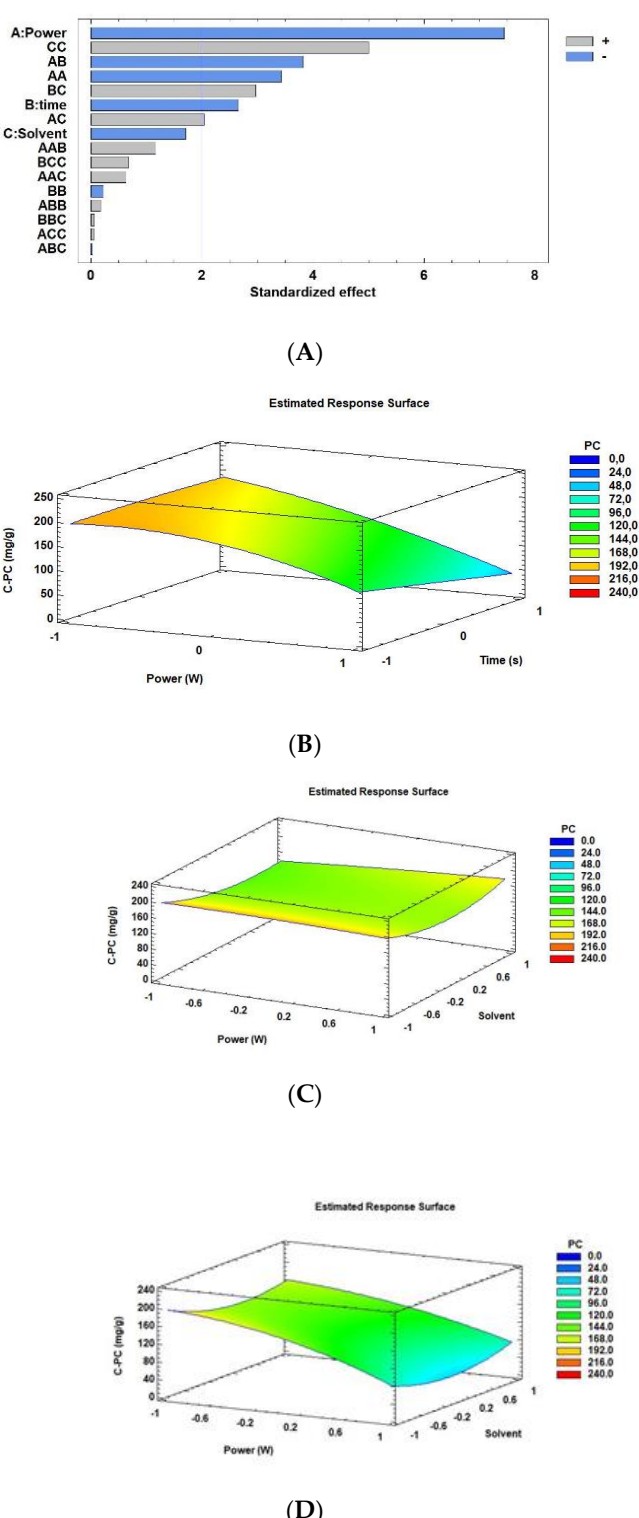

(**A**)

(**B**)

(**C**)

(**D**)

**Figure 1.** (**A**) Standardized Pareto charts for the C-PC content as response variable studied in the experimental design (grey and blue bars show negative and positive effects, respectively), and their corresponding response surfaces using the MW extraction technique. The figures represent the optimal variable responses such as: (**B**) solvent as distiller water; (**C**) power at 100 W; (**D**) time at 30 s. Data are shown as mean ± SD, n = 3.

**Table 3.** Regression coefficients (values of variables are specified in their original units) and statistics for the fit obtained by multiple linear regression.

| Terms of the Model | C-PC Content (mg/g) | |
| --- | --- | --- |
| | Estimate | *p*-Value |
| Constant | 120.021 | |
| A:Power | −51.8069 | 0.0000 * |
| B:time | −18.4825 | 0.0100 * |
| C:solvent | −11.8465 | 0.0938 |
| AA Power × power | −18.512 | 0.0011 * |
| AB Power × time | −14.5478 | 0.0003 * |
| AC Power × solvent | 7.76572 | 0.0459 * |
| BB time × time | −1.20219 | 0.8243 |
| BC time × solvent | 11.3066 | 0.0043 * |
| CC solvent × solvent | 26.9978 | 0.0000 * |
| AAB Power × power × time | 0.0 | 0.2485 |
| AAC Power × power × solvent | 0.0 | 0.5327 |
| ABB Power × time × time | 1.15228 | 0.8621 |
| ACC Power × solvent × solvent | 0.338278 | 0.9803 |
| BBC Time × time × solvent | 0.0 | 0.9593 |
| BCC Time × solvent × solvent | 4.42314 | 0.9513 |
| ABC Power × time × solvent | 12.1269 | 0.5056 |
| Statistics for goodness of fit of the model | | |
| $R^2$ | 0.84615 | |
| RSD | 22.885 | |
| P | 0.4008 | |
| RRSD (%) | 3.00351 | |

$R^2$-determination coefficient, RSD-residual standard deviation, *p*-value of the lack-of-fit test for the model, RRSD-residual standard deviation expressed as a percentage of the mean value of the response; * significant coefficients of the model.

For example, in two studies of fucoxanthin and phycoerythrin extraction with the MW process, Pasquet, et al. [30] and Juin, Chérouvrier, Thiéry, Gagez, Bérard, Joguet, Kaas, Cadoret and Picot [28] respectively, suggested that the increase of time or power would not have any impact on the pigment's extraction yield. It was also described by İlter, Akyıl, Demirel, Koç, Conk-Dalay and Kaymak-Ertekin [29], that the highest content of C-PC was provided with ultrasound extraction (US) followed by MW using different solvents with 102.98 ± 1.14 and 28.82 ± 1.10 mg/g, respectively.

This case was highlighted with 1.5% (*w/v*) aq, CaCl$_2$ as solution to C-PC extract with a value of 96.6 mg/g from *Arthrospira platensis.* Therefore this technique has been proposed as an efficient and rapid process to extract antioxidant compounds or pigments from plants, oils from vegetables, allowing reduced solvent consumption and shorter extraction times, with equivalent or higher extraction yields [30,31]. In particular, the technique by MW showed that it helped to accelerate the pigment extraction kinetics in diatoms by a few minutes and using a low volume of solvent as was studied by Pasquet, Chérouvrier, Farhat, Thiéry, Piot, Bérard, Kaas, Serive, Patrice and Cadoret [30]. Moreover, it has also been described as a useful technique to extract compounds from armored species or with an exopolysaccharide envelope produced by species as *Porphyridium* sp. where the bioactive compound extractions are troublesome [32]. Finally, the MW process was used because proteins are not degraded compared to other methods, due to their short extraction time [29].

*3.2. Optimization of HPH Extraction Conditions for C-PC of Arthrospira Maxima*

Optimization of the HPH extraction conditions is summarized in Table 2. As shown, higher C-PC content was obtained using Na-phosphate buffer as solvent and a pressure of 1400 bar (~272.4 mg/g as predicted optimal value and ~291.9 mg/g as real value). Figure 2 also represented standardized Pareto charts (Figure 2A) where non-significant variables were eliminated from the model and response

surfaces (Figure 2B). Secondly, the regression equation was also performed in the HPH method. It was adjusted to the C-PC content as the following model (2):

$$\text{C- PC (mg/g)} = 211.26 + 10.36 \cdot P + 10.96 \cdot S - 29.20 \cdot P2 + 8.37 \cdot P \cdot S + 47.22 \cdot S2 \tag{2}$$

where C-CP is C-phycocyanin content in mg/g, P: is pressure in the bar, and S: is the solvent type (Table 4).

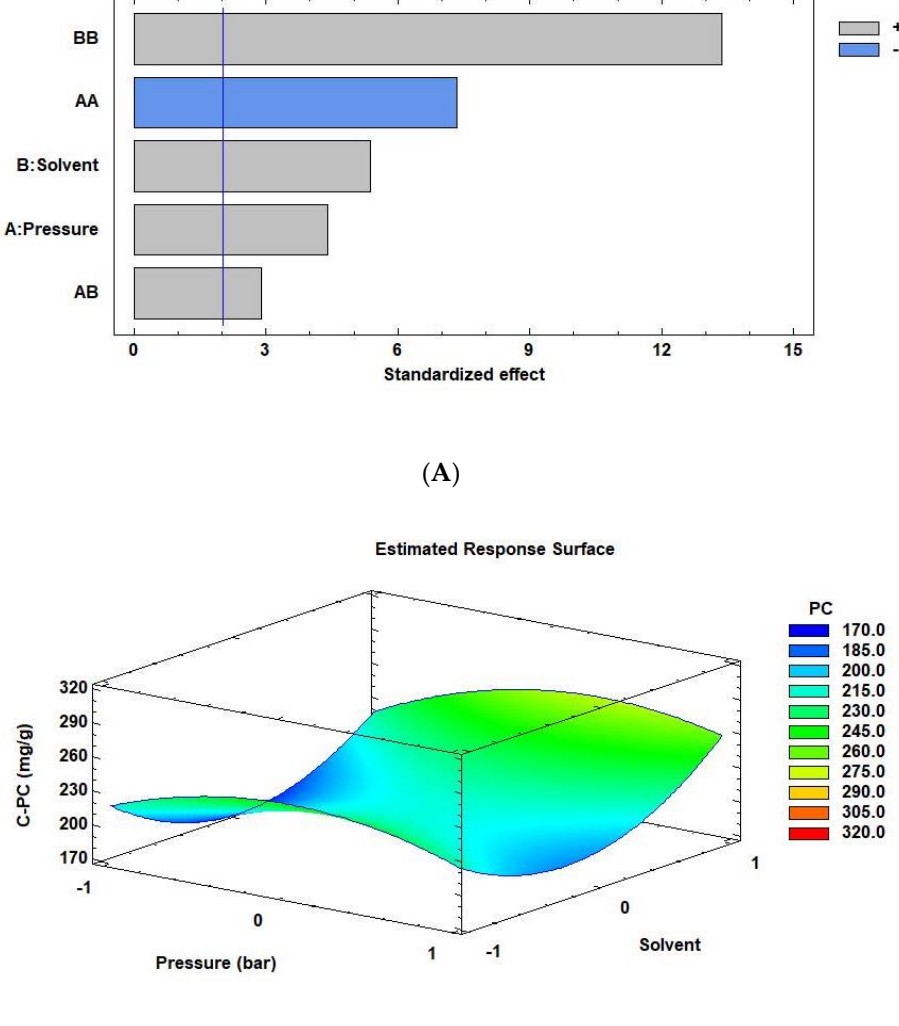

(**A**)

(**B**)

**Figure 2.** (**A**) Standardized Pareto charts for the C-PC content as response variable studied in the experimental design (grey and blue bars show negative and positive effects, respectively), and their corresponding (**B**) response surfaces using the HPH extraction technique. Data are shown as mean ± SD, n = 3.

Our results showed that after each disruption, the amounts of recovered C-PC from *A. maxima* increased in direct correlation with a higher operating pressure, which is confirmed by what is described by Halim, et al. [33]. However, we also observed that the increase from 1400 to 1600 bar caused a decrease in the C-PC content of *A. maxima*. Among the cell disruption methods, HPH is one of the most promising treatments for the complete disruption of biological cells [34,35]. Several studies report that the pressure increases the force with which the cells impact on the ring, inside valve seat, resulting in more efficient cell disruption [36]. In particular, the study performed by Carullo, Abera, Casazza, Donsì, Perego, Ferrari and Pataro [34] where they observed the effects of different HPH passes in the

cellular rupture of *Chlorella vulgaris* also were capable of inducing the complete disruption of the cells when they worked with higher pressures (to 1500 bar)

The supernatants obtained from HPH treated samples were characterized by a strong green color being the most intensive in all extraction processes similar to the results expose by Carullo, Abera, Casazza, Donsì, Perego, Ferrari and Pataro [34]. The major advantage in HPH is that this process can be scaled up easily with large volumes of biomass and with a wide range of disruption of cells (algae, bacteria and yeast). This technique is also non-selective with the product's recovery and with temperature increasing but it could be improved with the solvent selection and with the optimization of HPH passes, respectively [34,37,38].

In particular, chlorophyll extraction from *Spirulina platensis* using the HPH process was described by Choi and Lee [39]. Unlike our results, its study greatly increased the extraction yield of chlorophyll at less pressure than our data (improved at 690 bar and decreased at 1370 bar). Finally, the good extraction yield obtained in HPH allowed a relevant comparison of the processes' performances with other environmentally friendly techniques such as ultrasound, supercritical fluids, pressurized liquid extraction etc.

**Table 4.** Regression coefficients (values of variables are specified in their original units) and statistics for the fit obtained by multiple linear regression.

| | C-PC Content (mg/g) | |
|---|---|---|
| **Terms of the Model** | **Estimated** | ***p*-Value** |
| constant | 211.26 | |
| A Pressure | 10.3651 | 0.0001 * |
| B Solvent | 10.9577 | 0.0000 * |
| AA Pressure × Pressure | −29.2046 | 0.0000 * |
| AB Pressure × solvent | 8.37285 | 0.0061 * |
| BB Solvent × solvent | 47.2129 | 0.0000 * |
| Statistics for goodness of fit of the model | | |
| $R^2$ | 0.881058 | |
| RSD | 11.1752 | |
| P | 0.4979 | |
| RRSD (%) | −7.51706 | |

$R^2$-determination coefficient, RSD-residual standard deviation, *p*-value of the lack-of-fit test for the model, RRSD-residual standard deviation expressed as a percentage of the mean value of the response; * significant coefficients of the model.

### 3.3. Comparison of C-PC Content of Arthrospira Maxima between Conventional and Green Extraction Techniques

Two conventional methods were also carried out to extract C-PC from *A. maxima.*, Maceration (M) and freeze–thawing (FT) extraction, these being the conventional methods that were compared with green extraction techniques. Figure 3 shows C-PC content with the conventional extraction method using three independent solvents previously described. In this case, M extraction was the worst method to extract C-PC from *A. maxima* (~151.8 mg/g as average with any solvent). However, the FT method was better and the results were similar to that of MW extraction (Na-buffer phosphate ~ 232.8 mg/g). According to the literature, the FT method is commonly used to extract phycocyanin from wet biomass, due to the fact that it improves the extraction efficiency [40,41]. Although Silveira, et al. [42] reported that the extraction of phycocyanin with water did not differ much from that phosphate buffer, we found relevant significances in all methods except the M process. It could be because the buffer used in that study was 10 mM sodium acetate buffer (pH 5.0) different from our buffer used. The isoelectric point is essential to proteins interacting with the solvent, so pH is an important factor to improve the extraction.

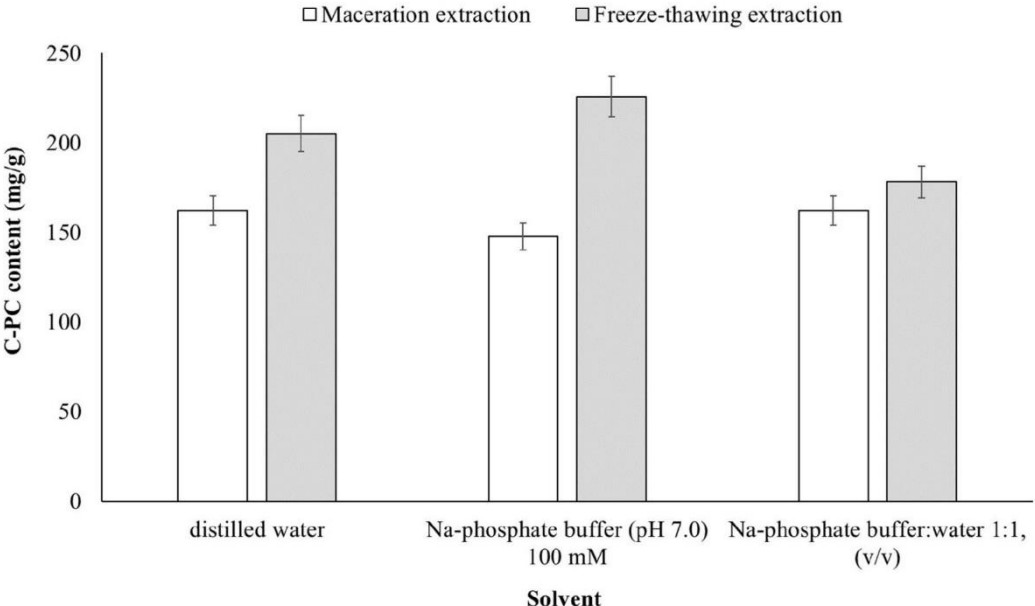

**Figure 3.** C-PC content by conventional extraction methods using three types of solvent. Data are shown as mean ± SD, n = 3.

Romero, et al. [43] has described in their research four different methods for the extraction of phycobiliproteins from *Arthrospira maxima*, recording that the highest content of phycocyanin recovery was 78.5 mg/L (equivalent to 8.7%, respect to dry biomass) under the phosphate regulator method with dry ice. This result was higher than that reached by Sandeep, et al. [44]. Sandeep et al. (2015) with *A. platensis* obtained a 5.0% in extraction yield, using phosphate buffer as the extraction solvent. In our work, using the FT method gave better results with Na-buffer phosphate ~ 232.8 mg/g (equivalent to 23.3% dry biomass). Even our lesser phycocyanin content (~15.0%) was higher than the results expressed by Becker [45] and Simpore, et al. [46] who indicated that in *Arthrospira* sp. the phycocyanin concentrations can reach between 9.0% and 15.0% of dry biomass. Finally, these conventional methods can be useful for small-scale extraction of bioactive compounds but they can be costly when the production scale is increased.

## 4. Conclusions

In this work, two green rapid extraction processes (MW and HPH) have been tested to study the optimal conditions to extract C-PC content from *A. maxima*. Moreover, conventional methods were also performed to compare the results of extraction. The higher results were under the HPH technique with 1400 bar and using Na-phosphate buffer as solvent, being 1.3-fold higher C-PC content than the MW and FT methods resulting from *A. maxima*. Moreover, M extraction proved to be the worst method to extract C-PC from *A. maxima* (~151.8 mg/g as average with any solvent).

It is important to highlight that the use of HPH and MW as green extraction technologies are appropriate for various reasons versus conventional methods such as: (i) they can be scaled up with major volume of biomass, (ii) they can disrupt species with a high mechanical resistance where the bioactive compounds' extraction are very difficult, and (iii) higher extraction yields are possible to obtain, thereby improving the costs of the process. In particular, our results with the MW process could be enhanced if this technique were combined with another green process such as ultrasound. The solvent selection is a restrictive factor in the industrial process. For this reason, water was chosen as the extractant, because it produced high C-PC concentration and it has a low cost. Finally, the use of experimental design based on RSM was confirmed as a useful statistical tool for the optimization of extraction conditions. The results demonstrated a high content of C-PC from *A. maxima* under green

extraction techniques with possible use in industrial applications. In addition, cyanobacteria was confirmed to be a good choice for its high phycocyanin content, easy cultivation, and processing.

**Author Contributions:** Conceptualization: M.C.R.-D. and P.C.; Data curation: E.M. and M.J.; Formal analysis: M.C.R.-D. and P.C., Funding acquisition: M.C.R.-D. and P.C.; Investigation: M.C.R.-D., E.M. and M.J.; M.C.R.-D., E.M. and M.J.; Methodology: M.C.R.-D. and P.C.; Resources: C.J.; Writing—original draft: M.C.R.-D. and M.J.; Writing—review and editing: P.C.

**Funding:** This research was funded by public funds of Chile, CONICYT (PAI-79160037 and FONDECYT-11170017 projects, respectively).

**Acknowledgments:** The group at Universidad de Antofagasta, Chile "Laboratorio de Microencapsulación de Compuestos bioactivos, LAMICBA" appreciate the support at the University of Antofagasta for "Development in Scientific Research of undergraduate" code 787-2018 and give special recognition to Carlos Dario Alfonso for his help in translation.

**Conflicts of Interest:** No conflicts, informed consent, human or animal rights applicable.

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
