# Peer review of "Rapid Green Extractions of C-Phycocyanin from Arthrospira maxima for Functional Applications"

_applsci, doi:10.3390/app9101987_

Round 1

Reviewer 1 Report

This research paper provides a brief but thorough insight into the extraction of C- C-Phycocyanin from Arthrospira maxima for different functional applications. The experimental method is solid and statistical fundamentals of data processing are used.

The short mention of the draw backs of the presented techniques, regarding costs, are necessary and very relevant for industrial adoption.

The paper should be checked again for minor errors, e.g. line 174 ”In this case, it was highlighted CaCl2 as solvent with a value of 96.6 mg/g from Arthrospira platensis.” It can be assumed that some kind of buffer is used for extraction, but the phrasing is unclear.

Unfortunately the paper is limited to laboratory observations and data presentation. While the presented data in itself is surely worthy of publication, additional insight into the theoretical background of mass transfer of the occurring effects would be interesting.

Author Response

Dear Reviewer 1,

We have responded to their questions (attached manuscript) where you can see in red color the modifications suggested.

Anyway, you will see in italics below the answer:

1. The short mention of the draw backs of the presented techniques, regarding costs, are necessary and very relevant for industrial adoption.

In paragraph 59-61 is added this information and it was supplement with paragraph 65-70, 87-89, 252-258, and Conclusion section

2. The paper should be checked again for minor errors, e.g. line 174 ”In this case, it was highlighted CaCl2 as solvent with a value of 96.6 mg/g from Arthrospira platensis.” It can be assumed that some kind of buffer is used for extraction, but the phrasing is unclear.

Yes, this solution CaCl2 1.5 %w/v was used as extractant (you could see line 207-208)

3. Unfortunately the paper is limited to laboratory observations and data presentation. While the presented data in itself is surely worthy of publication, additional insight into the theoretical background of mass transfer of the occurring effects would be interesting.

We have improved the discussion, it could see in red color in the manuscript added.

Moreover, we have purposed another full title if you consider the change and we have enhanced the English redaction. I hope you consider accepting the manuscript in the journal.

Thanks for all. 

Best regards,

Mari Carmen Ruiz-Dominguez

Reviewer 2 Report

1). Abstract - please make it more interesting - current version gives an impression that whole manuscript is a simple optimization of extraction process, no science, just routine work

2). Introduction - please emphasize more precisely the need to produce this particular compound - add few more sentences. Highlight the need to conduct more research in this field.

3). Discussion is not very insightful - some issues should discussed in details. Please refer to other reports. Is your method more universal in comparision to others ? What about rules of green chemistry ? Does it follow them ? Please highlight novelty behind this study. Most of the extraction method tend to extract undesired compounds - is this a case in your research too ? How pure is the extract ? Sometimes lower yield results in a better product quality. Please refer to those issues in the discussion.

4). Conclusions should be more universal - please make your manuscript interesting for wide range of scientists.

Author Response

Dear Reviewer 2,

We have attached the manuscript revised (you can see in red color the modifications suggested). Anyway, we answer below in italics their comments:

1). Abstract - please make it more interesting - current version gives an impression that whole manuscript is a simple optimization of extraction process, no science, just routine work

We have greatly modified the abstract and the focus is different. 

2). Introduction - please emphasize more precisely the need to produce this particular compound - add few more sentences. Highlight the need to conduct more research in this field.

We have modified the section and you can see in paragraphs 59-61, 65-70 and 87-89 the suggestions. 

3). Discussion is not very insightful - some issues should discussed in details. Please refer to other reports. Is your method more universal in comparision to others ? What about rules of green chemistry ? Does it follow them ? Please highlight novelty behind this study. Most of the extraction method tend to extract undesired compounds - is this a case in your research too ? How pure is the extract ? Sometimes lower yield results in a better product quality. Please refer to those issues in the discussion.

In this question, we have intensified the discussion with more details and references. We have highlighted the benefits of using green extraction techniques versus conventional from several points of view.  Moreover, we have intensified the role of C-PC industrially and their possible used. All you can see in red color in the manuscript revised.

4). Conclusions should be more universal - please make your manuscript interesting for wide range of scientists.

We have modified this section. Now it is more general and with a view more scientistic. 

Moreover, we have again improved the English redaction with an English colleague. Finally, we suggested another full title if you considered the change. 

Thanks.

Best regards,

Mari Carmen Ruiz-Domìnguez
